# Detection of Alpha- and Betacoronaviruses in Small Mammals in Western Yunnan Province, China

**DOI:** 10.3390/v15091965

**Published:** 2023-09-20

**Authors:** Fen-Hui Xu, Pei-Yu Han, Jia-Wei Tian, Li-Dong Zong, Hong-Min Yin, Jun-Ying Zhao, Ze Yang, Wei Kong, Xing-Yi Ge, Yun-Zhi Zhang

**Affiliations:** 1School of Public Health, Institute of Preventive Medicine, Dali University, Dali 671000, China; xufenhui1569@163.com (F.-H.X.); hanpeiyu1511@gmail.com (P.-Y.H.); tjwfatorange@outlook.com (J.-W.T.); zld2019106326@163.com (L.-D.Z.); 13099916216@163.com (H.-M.Y.); zhaojunying0714@163.com (J.-Y.Z.); yangzeyz@163.com (Z.Y.); kongwei4357@163.com (W.K.); 2Key Laboratory of Pathogen Resistant Plant Resources Screening Research in Western Yunnan, Dali 671000, China; 3Key Laboratory of Cross-Border Prevention and Control and Quarantine of Zoonotic Diseases in Yunnan, Dali 671000, China; 4College of Biology & Hunan Provincial Key Laboratory of Medical Virology, Hunan University, Changsha 410012, China; xyge@hnu.edu.cn

**Keywords:** small mammalian coronavirus, quantitative reverse transcription polymerase chain reaction (qRT-PCR), tissue tropism, cross-species transmission

## Abstract

The genetic diversity of coronaviruses (CoVs) is high, and their infection in animals has not yet been fully revealed. By RT-PCR detection of the partial RNA-dependent RNA polymerase (RdRp) gene of CoVs, we screened a total of 502 small mammals in the Dali and Nujiang prefectures of Western Yunnan Province, China. The number of overall CoV positives was 20, including β-CoV (*n* = 13) and α-CoV (*n* = 7), with a 3.98% prevalence in rectal tissue samples. The identity of the partial RdRp genes obtained for 13 strains of β-CoV was 83.42–99.23% at the nucleotide level, and it is worth noting that the two strains from Kachin red-backed voles showed high identity to BOV-36/IND/2015 from Indian bovines and DcCoV-HKU23 from dromedary camels (*Camelus dromedarius*) in Morocco; the nucleotide identity was between 97.86 and 98.33%. Similarly, the identity of the seven strains of α-CoV among the partial RdRp sequences was 94.00–99.18% at nucleotide levels. The viral load in different tissues was measured by quantitative RT-PCR (qRT-PCR). The average CoV viral load in small mammalian rectal tissue was 1.35 × 10^6^ copies/g; differently, the mean CoV viral load in liver, heart, lung, spleen, and kidney tissue was from 0.97 × 10^3^ to 3.95 × 10^3^ copies/g, which revealed that CoV has extensive tropism in rectal tissue in small mammals (*p* < 0.0001). These results revealed the genetic diversity, epidemiology, and infective tropism of α-CoV and β-CoV in small mammals from Dali and Nujiang, which deepens the comprehension of the retention and infection of coronavirus in natural hosts.

## 1. Introduction

Coronaviruses (CoVs) belong to Orthocoronavirinae subfamily, *Coronaviridae* family in the *Nidovirales* order. CoV is an enveloped, positive-sense, single-stranded RNA virus with a genome length of around 30,000 bp [1]. According to the International Committee on Taxonomy of Viruses (ICTV) classification criteria, based on the five structural domains of CoV polyprotein 1ab (pp1ab) (i.e., RNA-dependent RNA polymerase (RdRp), Nidovirus RdRp-associated nucleotidyltransferase (NiRAN), 3-chymotrypsin-likepro-tease (3CL-pro), helicase of superfamily 1 (HEL1), and zinc-binding domain (ZBD)), the subfamily Orthocoronavirinae was classified into four genera (52 species and 26 subgenera): *Alphacoronavirus* (α-CoV) including 26 species and 15 subgenera, *Betacoronavirus* (β-CoV) including 14 species and 5 subgenera, *Deltacoronavirus* (δ-CoV) including 7 species and 3 subgenera, and *Gammacoronavirus* (γ-CoV) including 5 species and 3 subgenera (https://ictv.global.taxonomy, accessed on 19 August 2023). CoVs show host specificity and tissue infection preference. Typically, α-CoV and β-CoV infect mammals; γ-CoV and δ-CoV mainly infect birds, and some can also infect mammals [2]. Since Hamre et al. [3] discovered human coronavirus 229E (HCoV-229E) in the United States in 1966, a total of seven human coronaviruses (HCoVs) have been identified; besides HCoV-229E, the remaining six human coronaviruses are NL63 (HCoV-NL63), OC43 (HCoV-OC43), HKU1 (HCoV-HKU1), severe acute respiratory syndrome coronavirus (SARS-CoV), Middle East respiratory syndrome coronavirus (MERS-CoV), and severe acute respiratory syndrome coronavirus 2 (SARS-CoV-2). According to known knowledge, all HCoVs have been found to have related prototypes in animals: HCoV-OC43- and HKU1-related strains in rodents [4] and CoVs similar to SARS-CoV, MERS-CoV, HCoV-NL63, HCoV-229E, and SARS-CoV-2 in bats [5,6]. Except for SARS-CoV, MERS-CoV, and HCoV-OC43, whose intermediate hosts may be civets [7,8,9], dromedaries [10], and cattle [11], respectively, it is not well understood how the remaining four CoVs are transmitted to humans. The high mutation characteristics of CoV present a high degree of genetic diversity while promoting its transmission.

CoVs are widespread in nature, which plays an important position in emerging infectious diseases. Zoonotic CoV was discovered in the 1960s [12], and the frequency and scope of influence are increasing. The vast majority of animal infectious diseases originate from wildlife, and rodents represented by rats carry a variety of zoonotic pathogens, which play an extremely important role in emerging zoonoses. In many places, rodents come into close contact with humans, farm animals, or pets. Rodents in towns and around cities provide a nexus between wildlife communities and humans, exposing humans to some zoonoses circulating in these natural ecosystems [13]. The mouse coronavirus has been associated with rodents, and the prototype virus first named murine hepatitis virus (MHV) and then renamed *Murine coronavirus* by the ICTV was the first isolated in mice in 1949 [14]. In 1970, a variant called the rat coronavirus (RCV) was discovered in rats [15]. It was not until 2014 that a new murine coronavirus, *Lucheng Rn rat coronavirus* (LRNV) from the Norway rat (*Rattus norvegicus*) was discovered, along with two new variants of β-CoV—*Longquan Aa mouse coronavirus* (LAMV) from the striped field mouse (*Apodoses agrarius*) and *Longquan Rl rat coronavirus* (LRLV) from the lesser rice field rat (*Rattus Lossea*) [16]. The CoVs found in rodents are divided into two lineages: the A lineage of β-CoV and a separate lineage of α-CoV [17]. Then, in 2015, a new murine coronavirus, *China Rattus HKU24*, belonging to the A lineage of β-CoV was discovered in Norway rats in China [4]. In 2022, a novel coronavirus, *Myodes coronavirus 2JL14*, was reported in Swedish bank voles (*Myodes glareolus*) [18]. Recently, two new CoVs, namely *Suncus murinus coronavirus X74* from Asian house shrews and *Sorex araneus coronavirus T14* from common shrews, were discovered [19,20]. In Yunnan Province of China, we have detected genetically diverse α-CoV and β-CoV in a variety of rats, such as Chevrieri’s field mouse (*Apodemus chevrieri*) and large Chinese vole (*Eothenomys miletus*), and identified the CoV genomes [21]. All these findings suggest that small mammals, especially rodents, may carry a wide range of CoVs.

In this study, we collected a total of 502 small mammals belonging to 18 species in 12 genera and 4 orders in Yunnan Province of China and investigated the genetic diversity and epidemic of CoV by RT-PCR screening and sequencing of the partial RdRp gene of CoV. The qRT-PCR method was further established for quantification of CoV, and the viral copy number of CoV was quantified, as well as tissue tropism of CoV in heart, liver, spleen, lung, kidney, and rectal tissues of RT-PCR positive samples. The results of this study increase our understanding of CoV diversity and provide a method for rapid, accurate, and reliable screening of CoV.

## 2. Materials and Methods

### 2.1. Ethics Statement

The collection of small animals was performed by veterinarians with approval from the Animal Ethics Committee of Dali University (DLDXLL2020007).

### 2.2. Sample Collection and Processing

Small mammal samples were collected from August 2020 to August 2022 in residential areas, arable areas, and wild bush areas in Heqing County, Dali City of Dali Prefecture, and Gongshan County, Lushui City of Nujiang Prefecture, Yunnan Province, China (Figure 1), using freshly fried fritters as bait. The sample trapping was performed using cage-type traps. The collected small mammals were brought back to the laboratory and euthanized, with species identification initially based on morphology, followed by further molecular identification of the species by sequence analysis of the mitochondrial (mt)-cytochrome b (*Cytb*) gene [22]. The samples were dissected in a sterile environment, and the heart, liver, spleen, lung, kidney, and rectal tissues were collected in 2 mL cryogenic vials (CORNING, Shanghai, China) and stored temporarily in liquid nitrogen. The samples were then stored at −80 °C before further laboratory analyses.

### 2.3. DNA and RNA Extraction

Under aseptic conditions, approximately 1 g of rectal and other tissue samples was cut into GeneReady Animal PIII crushing tubes (Life Real, Hangzhou, China), and 600 μL of sterilized phosphate-buffered saline (PBS) was added, followed by grinding in a GeneReady Ultimate grinder (Life Real, Hangzhou, China). Then, 300 μL of the supernatant of the ground tissue sample was added to the nucleic acid extraction or purification kit (MagaBio plus Virus DNA/RNA Purification Kit Ⅲ, Hangzhou, China), and the sample DNA/RNA was extracted in a fully automated nucleic acid extraction and purification instrument (BIOER, Hangzhou, China) according to the instructions, dispensed, and stored at −80 °C until further analysis.

### 2.4. Primary Screening of CoV and Amplification of Partial RdRp Fragments

Semi-nested PCR (RT-PCR) amplification was used for the conserved regions of the RdRp gene of CoV (Table 1) [23]. The primers were synthesized by Shanghai Sangon Biotech, and the reaction system was 25 μL for both rounds. The first round of RT-PCR was performed using the Fastking One Step RT-PCR kit (TIANGEN, Beijing, China). The reaction system of the first round was as follows: 2 × Fasting One Step RT-PCR MasterMix 12.5 μL, 25× RT-PCR Enzyme Mix 1 μL, CoV-FWD3 and CoV-FWD4/other (20 μM) 1 μL each, RNase-Free ddH_2_O 6.5 μL, RNA template 3 μL. The first-round PCR conditions were as follows: 30 min reverse transcription at 42 °C, followed by 35 cycles at 95 °C, 3 min pre-denaturation at 94 °C, 30 s denaturation at 48 °C, 30 s 35 annealing at 72 °C, 30 s elongation, and a final extension 72 °C for 5 min and 1 min cooling at 10 °C. The second round of PCR was performed with 2× Phanta Max Master Mix (Vazyme, Nanjing, China), and the reaction system was as follows: 2 × Phanta Max Master Mix 12.5 μL, 1 μL each of CoV-RVS3 and CoV-FWD4/other (20 μM), 9.5 μL of RNase-Free ddH_2_O, and 1 μL of the product of the first-round PCR as the template. The second-round PCR conditions were as follows: 95 °C, 3 min pre-denaturation, 35 cycles at 94 °C, 15 s denaturation at 50 °C, 15 s annealing at 72 °C, 30 s extension, and a final extension of 72 °C for 5 min and 1 min cooling at 10 °C. The length of the amplification product after two rounds of RT-PCR was approximately 443 bp. The second-round RT-PCR products were identified by agarose gel electrophoresis, and the positive amplification products that matched the expected size were purified by gel cutting (OMEGA Bio-tek, Norcross, GA, USA) and sent to Sangon Biotech for bi-directional sequence determination. For well-sequenced positive samples, specific primers were designed to amplify part of the ORF1ab fragment by multiple sequence alignment with the published coronavirus genome. To exclude PCR contamination, positive samples were verified by two independent PCRs performed by two different experimenters.

### 2.5. Virus Sequence Identification and Phylogenetic Analysis

Sequences were assembled by the DNAstar Lasergene 7.1.0 software package and manually edited and cut to generate the final sequence of the viral gene. Similarity matching analysis was performed using the National Center for Biotechnology Information (NCBI) online Basic Local Alignment Search Tool (BLAST)-based search tool. The CoV reference sequence set representing the RdRp gene was downloaded from GenBank, and sequence comparison was performed using ClustalX2. All viral sequences were constructed using the maximum likelihood method in MEGAX 11.0 with a total of 1000 bootstrap replicates for generation. The evolutionary distances were computed using the Kimura 2-parameter method, in which a self-spread value greater than 70% is generally considered a reliable evolutionary branching, and visualized in iTOL (https://itol.embl.de/, accessed on 20 May 2023). The same method was used to construct the evolutionary tree based on the mt-*Cytb* gene for the corresponding small mammalian hosts. The α-CoV and β-CoV sequences in this study were deposited to GenBank under the following numbers: OR223161-OR223180. The mt-*Cytb* gene sequences from small mammals in this study were deposited to GenBank under the following numbers: OR223181-OR223200.

### 2.6. Construction of Plasmids and Determination of Virus Copies

The partial ORF1ab fragments of the representative α-CoV and β-CoV strains were cloned into the pEASY-T1 vector (TransGen Biotech, Beijing, China), and the T-loaded products were transformed into DH5α *E. coli* cells. The inserted target genes were confirmed by sequencing after bacteriophage amplification. Small amounts of plasmids were extracted using the Plasmid Mini Kit Ⅰ (Omega Bio-tek, Norcross, GA, USA) and stored at −80 °C in separate devices. Then, the concentration of the extracted coronavirus plasmid was determined using an ultraviolet spectrophotometer (Life Real, Hangzhou, China) after gradual thawing, and then according to equation (1), the plasmid concentration was converted to copies for the establishment of standard curves and quantitative analysis as a positive control.
copies/μL = plasmid concentration (ng/µL) × 10^−9^ × 6.02 × 10^23^/(660 × DNA length)(1)

### 2.7. Primer and TaqMan Probe Design and Optimization

After part of the amplified ORF1ab fragment was aligned with the existing reference sequence in GeneBank, different specific primers and TaqMan probes were designed for the conserved sequences within the ORF1ab gene of α-CoV and β-CoV (Table 1). Positive standards were diluted with RNase-Free ddH_2_O in a 10-fold gradient and then used as templates for condition optimization and stored at −20 °C. The primer and TaqMan probe concentrations and annealing temperature within the qRT-PCR system were optimized by several experiments to determine the optimal amplification conditions and reaction system.

HiScript^®^ Ⅱ U+ One Step qRT-PCR Probe Kits (Vazyme, Nanjing, China) were used. After several experiments to optimize the conditions, the reaction system for qRT-PCR was as follows: 2 × One Step U+ Mix 10 μL, One Step U+ Enzyme Mix 1 μL, 50 × ROX Reference Dye (2) 0.4 μL, primers for both α-CoV and β-CoV 0.4 μL (10 μM), TaqMan probe 0.2 μL (10 μM), RNase-Free ddH_2_O 5.6 μL, and RNA template 2 μL. The reaction system for the assay was 20 μL, and the amplification reaction was performed using the Applied Biosystems 7500 Real-Time PCR system (Thermo Fisher Scientific, Waltham, MA, USA). The amplification conditions were as follows: 55 °C, 15 min reverse transcription, 45 cycles at 95 °C, 30 s pre-denaturation, 95 °C 10 s denaturation, α-CoV and β-CoV annealing temperatures and fluorescence signal acquisition times of 51 °C, 34 s, and 60 °C, 34 s, respectively.

### 2.8. Establishment of Standard Curves

The positive standards diluted in a 10-fold gradient were used as templates; each concentration was repeated three times, and the average of the three times was taken. The standard curve was plotted using the logarithm of the copies of the positive standard as the horizontal coordinate and the cycle threshold (Ct) value corresponding to the assay as the vertical coordinate, and the slope and correlation coefficient were calculated.

### 2.9. Evaluation of qRT-PCR Methods

#### 2.9.1. Sensitivity

The limit of detection for qRT-PCR is determined by detecting a positive standard of serial dilution. α-CoV and β-CoV plasmid DNA concentrations were 356.828 ng/µL and 312.525 ng/µL, respectively, and the concentrations before dilution of the two genera were 6.09 × 10^10^ copies/µL and 5.24 × 10^10^ copies/µL, respectively. The plasmids were serially diluted 10-fold using RNase-Free ddH_2_O and used as qRT-PCR templates.

#### 2.9.2. Specificity

To assess the specificity of qRT-PCR, Hantavirus, *Orientia tsutsugamushi*, and Hepatitis E virus provided by our laboratory were compared with CoV.

#### 2.9.3. Repeatability and Stability

Six concentration gradients (1.00 × 10^4^–1.00 × 10^9^ copies/µL) of positive standards diluted in a 10-fold gradient were used as templates, and each concentration was repeated 3 times as intra-group replicates, and the above operation was performed once a week for a total of 3 times as inter-group replicates. RNase-Free ddH_2_O was used as a negative control group for testing the intra-group and inter-group variation of different concentrations of positive standards. The mean Ct value (Mean Ct), standard deviation (SD), and coefficient of variation (CV) were calculated to evaluate the reproducibility and stability of qRT-PCR.

#### 2.9.4. Comparison of qRT-PCR and RT-PCR Assays

RT-PCR reactions were performed using serial 10-fold gradient dilutions of positive standards (1.00 × 10^9^–1.00 × 10^1^ copies/µL) as positive templates and RNase-free H_2_O as negative controls. The amplified PCR products were subjected to agarose gel electrophoresis to compare the sensitivity of qRT-PCR and RT-PCR methods.

### 2.10. qRT-PCR for Small Mammalian Samples

After CoV RNA extraction from tissues of small mammals captured in Dali and Nujiang prefectures of Yunnan from August 2020 to August 2022 according to the above method, all samples were tested for α-CoV and β-CoV according to the optimized qRT-PCR experimental conditions.

### 2.11. Tissue Tropism of Small Mammalian CoV

The samples testing positive by qRT-PCR were extracted from the heart, liver, spleen, lung, and kidney tissues according to the requirements of the nucleic acid extraction kit and instruments, and then the tissues of small mammalian samples with α-CoV and β-CoV positives were quantitatively analyzed to study their tissue tropism.

### 2.12. Statistical Analysis

Data were analyzed using a one-way ANOVA with GraphPad Prism software version 8.0 (GraphPad Software, San Diego, CA, USA). All results are expressed as the mean ± standard error of the mean (SEM). *p*-values < 0.05 were considered statistically significant, while *p*-values < 0.001 (three-star sign) and 0.0001 (four-star sign) were considered highly significant.

## 3. Results

### 3.1. Collection of Samples and Detection of CoV

A total of 502 small mammals of 18 species in 12 genera and 4 orders were collected in residential areas, arable areas, and wild bush areas in Dali and Nujiang prefectures of Yunnan Province (Table 2). The small mammals collected were healthy. RT-PCR was used to detect α-CoV and β-CoV RNA based on partial RdRp sequences. The number of overall positives for CoV was 20, including β-CoV (*n* = 13) and α-CoV (*n* = 7), with a 3.98% prevalence in rectal tissue samples. The collection dates and habitats of the 20 positive samples are shown in Appendix A. The prevalence of β-CoV was 3.54% (4/113) and 6.67% (6/90) in Chevrieri’s field mouse (*Apodemus chevrieri*) and Lancangjiang field mouse (*A. ilex*), respectively. The prevalence of both β-CoV and α-CoV in Kachin red-backed vole (*Eothenomys cachinus*) and Norway rat (*Rattus norvegicus*) was 2.22% (2/90) and 3.85% (1/26), respectively. The prevalence of α-CoV in White-footed Indochinese rat (*R. nitidus*) and long-tailed red-toothed shrew (*Episoriculus leucops*) was 75% (3/4) and 5.88% (1/17), respectively (Table 2).

### 3.2. Comparison of Partial RdRp Gene

The amplified partial RdRp sequences of the 13 strains of β-CoV in this study shared 83.42–99.23% nucleotide (nt) identity and 90.14–100.00% amino acid (aa) identity (Appendix A). It is worth noting that CoVNJ99 and CoVNJ142 from *E. cachinus* had the highest identity with BOV-36/IND/2015 from Indian bovines and DcCoV-HKU23 from dromedary camels (*Camelus dromedaries*) in Morocco, with the nt identity of 97.86–98.33%, indicating that their relationship is worth delving into The identity of nt and aa levels of the seven α-CoV positive sequences was 94.00–99.18% and 94.44–99.31%, respectively (Appendix A).

### 3.3. Phylogenetic Analysis

Among the 20 strains of CoVs identified in this study, CoVDL140 from *R. norvegicus*; CoVDL55, CoVDL75, CoVDL161, and CoVDL172 from *A. chevrieri*; and CoVNJ21, CoVNJ33, CoVNJ53, and CoVNJ55 from *A. ilex* were clustered with the *China Rattus HKU24* representative strains Ruili-874, Lijiang-41, and Lijiang-53, respectively, belonging to the *Embecovirus* subgenus of β-CoV. CoVNJ16 and CoVNJ56 from the *A. ilex* were more closely related to RtAp/SAX2015; CoVNJ99 and CoVNJ142 from the *E. cachinus* clustered together with BOV-36/IND/2015 from Indian bovines and DcCoV-HKU23 from dromedary camels in Morocco. Through the phylogenetic tree, it was shown that HCoV-OC43 is also closely related to CoVNJ99 and CoVNJ142. CoVNJ3 and CoVNJ52 from *E. cachinus*; CoVDL82 from *R. norvegicus*; CoVNJ135 from *Ep. Leucops*; and CoVNJ195, CoVNJ196, and CoVNJ207 from *R. nitidus* are more closely related to RtRl/FJ2015 and RtClan/GZ2015, all of which are α-CoVs, and viruses with high identity were found in insectivores and rodents. In addition, identical viruses can be identified in multiple species from the same geographical location; for example, CoVNJ135 from Ep. leucops and CoVNJ195, CoVNJ196, and CoVNJ207 from *R. nitidus* in Nujiang Prefecture are the most similar to RtRl/FJ2015 (Figure 2).

### 3.4. Establishment of qRT-PCR Standard Curves

The six consecutive dilution gradients of α-CoV and β-CoV standards (1.00 × 10^4^–1.00 × 10^9^ copies/μL) were selected as the log values of copy number on the X axis, and the obtained Ct values were plotted as the standard curve on the Y axis (Figure 3). This shows that the template of the gradient dilution has a good linear relationship with the Ct value.

### 3.5. Evaluation of qRT-PCR Methods

#### 3.5.1. Evaluation of Sensitivity

The experimental results showed that the minimum copy number detectable by positive standards for both α-CoV and β-CoV was 1.00 × 10^1^ copies/µL, indicating the good sensitivity of the established qRT-PCR method (Figure 4).

#### 3.5.2. Evaluation of Specificity

The results of the experiments showed that the specific primers designed only showed amplification curves for the positive standards of α-CoV and β-CoV and showed no amplification curves and no fluorescence signal in the negative control group, suggesting that the established qRT-PCR method had good specificity.

#### 3.5.3. Evaluation of Repeatability and Stability

Intra-group repeatability test: The experimental results showed that the SD in the standard group for each concentration of α-CoV was between 0.06 and 0.30, and the CV was between 0.26 and 1.09. The SD in the standard group for each concentration of β-CoV was between 0.05 and 0.39, and the CV was between 0.16 and 1.66 (Appendix A).

Inter-group repeatability test: The experimental results showed that the SD in the standard group for each concentration of α-CoV was between 0.01 and 0.16, and CV was between 0.03 and 0.96. The SD in the standard group for each concentration of β-CoV was between 0.04 and 0.47, and the CV was between 0.22 and 1.77 (Appendix A). All these data indicated that the established qRT-PCR method has good reproducibility and stability.

### 3.6. Comparison of qRT-PCR and RT-PCR Sensitivity

The minimum copy number detected by qRT-PCR for both α-CoV and β-CoV positive standards was 1.00 × 10^1^ copies/µL, while the minimum copy number detected by RT-PCR for α-CoV and β-CoV positive standards was 1.00 × 10^3^ copies/µL and 1.00 × 10^4^ copies/µL, respectively; these values were 10^2^ and 10^3^ times higher than those of RT-PCR, respectively (Figure 5).

### 3.7. Tissue Tropism

A total of 20 CoV-positive samples from Dali and Nujiang prefectures of Yunnan Province and CoV RNA naturally infected in the heart, liver, spleen, lung, kidney, and rectal tissues of the positive samples were quantified using the qRT-PCR method established in this study. The mean CoV copy number was 1.35 × 10^6^ copies/g in the rectum for all positive samples with the highest CoV copy numbers. The mean CoV copy numbers in liver, heart, spleen, lung, and kidney tissues were 3.95 × 10^3^ copies/g, 2.96 × 10^3^ copies/g, 2.93 × 10^3^ copies/g, 1.68 × 10^3^ copies/g, and 0.97 × 10^3^ copies/g, respectively (Table 3). The rectal tissue contained significantly higher viral copy numbers than the liver, heart, spleen, lung, and kidney tissues (*p* < 0.0001) (Figure 6). The remaining tissues except the rectal tissue also contained unequal copies of the virus, but there was no significant difference among tissues (*p* > 0.05).

## 4. Discussion

In this study, CoVs were detected in six species: *A. chevrieri*, *A. ilex*, *E. cachinus*, *R. norvegicus*, *R. nitidus*, and *Ep. leucops*. Among these hosts, β-CoV was found in three species, namely *A. chevrieri*, *A. ilex*, and *R. norvegicus*; α-CoV was found in two species, namely *R. nitidus*, and *Ep. leucops*; and α-CoV and β-CoV co-infection was found in two species, namely *E. cachinus* and *R. norvegicus*. These results showed that CoVs are widely present and highly diverse in Rodentia and Insectivora hosts. Other genetic characteristics of the α-CoV and β-CoV detected here need further genomic sequencing analysis. *A. chevrieri* and *A. ilex* were infected with highly similar CoVs and were from Dali and Nujiang prefectures in Yunnan Province, respectively. Nujiang Prefecture is located on the western border of Yunnan Province, China, adjacent to Myanmar and connected to Dali Prefecture in the southeast, which suggests a co-evolutionary relationship among CoVs and host animals. From the α-CoV and β-CoV and host co-evolutionary tree, it was found that CoVs of different genera could infect the same rodents and CoVs of the same genera could infect different rodents, suggesting the existence of cross-species transmission of α-CoV and β-CoV carried by rodents inhabiting the same habitat. It is speculated that this is the result of long-term evolution, mutual adaptation, and natural selection between CoVs and host animals.

CoVs can infect a wide range of host animals [24], and the cross-species transmission of CoVs has caused multiple epidemics of infection and disease in animals and humans, which have seriously affected human productive life and public health. Notably, this study detected CoVNJ99 and CoVNJ142 in *E. cachinus*; CoVNJ99 and CoVNJ142 are associated with CoVs carried by dromedary camel and bovines, which are intermediate hosts for the Middle East respiratory syndrome coronavirus. MERS-CoV is repeatedly detected in dromedary camels, and MERS-CoV isolated from dromedary camels is genetically and phenotypically similar to the CoVs that infect humans, including the virus spike protein, suggesting that CoVs in dromedary camels may be transmitted to humans [25]. True primary zoonotic infection is difficult to identify, and it is possible that the host or other intermediate host will be altered during the transmission of infection to humans [26]. However, to date, no investigations have been conducted on the presence of MERS-CoV or similar viruses in wild rodents [27]. Bovines are often grazed by herders in Nujiang Prefecture, so it is speculated that rats may have been exposed to bovine feces and there is cross-species transmission such that highly related CoVs from bovine infections can be detected in rats. The emergence and evolution of CoVs in new hosts are caused by a variety of factors, such as recombination, horizontal transfer of genes, gene duplication, and the shifting of open reading frames, all of which accelerate their infection with new hosts and enhance their ability to adapt to new hosts [28]. Although only partial fragments were obtained from CoVNJ99 and CoVNJ142 in this experiment, this CoV was detected in rats in wild bush areas in Nujiang Prefecture, Yunnan Province, China, and the related research should be strengthened. In the next step of research, our team will conduct a serological survey of the population in Gongshan County, Nujiang Prefecture, detect the full-length genome using next-generation sequencing (NGS), and isolate CoVNJ99 and CoVNJ142 strains.

The qRT-PCR established in this study had good sensitivity, specificity, stability, and reproducibility, and the highest CoV copy number (*p* < 0.0001) was found to be contained in small mammalian rectal tissue in quantitative studies, revealing that CoVs infecting small mammals have intestinal tropism. In the relevant research on SARS-CoV-2, some researchers have found that the feces remained positive in 23% of patients even after respiratory specimens tested negative for viral RNA; it is speculated that the gastrointestinal tract may be a specific target organ of the virus [29]. In previous studies on SARS-CoV, it was found that prolonged fecal shedding of viral RNA is common, and fecal samples remain positive even after the respiratory and/or sputum samples exhibit no detectable virus [30]. There were also traces of the highest average CoV load in rectal tissue in this study. However, unequal amounts of CoV copies were also detected in liver, heart, spleen, lung, and kidney tissues, with mean values ranging from 0.97 × 10^3^ to 3.95 × 10^3^ copies/g, suggesting that CoVs have a wide range of tissue tropism and may be transmitted by oral-fecal, urinary, and respiratory routes, which also provides evidence for the CoV transmission route. This also re-confirms that CoVs are respiratory, intestinal, hepatic, and renal pathogens in animals and humans and provides evidence for the clinical signs and symptoms of infected patients including respiratory, intestinal, hepatic, and renal manifestations, as well as other forms of disease [31]. In previous decades of research, different tissue orientations of rodent CoVs have been observed, with different MHV strains serving as prototypes of rodent CoVs that can infect variant tissues, with the MHV-A59 strain being predominantly hepatophilic and the MHV-JHM strain being predominantly neurotropic [32,33,34,35]. Rodent coronavirus (RCoV) and sialodacryoadenitis virus (SDAV) both primarily infect the respiratory tract [36]. In the present study, highly similar strains to HKU24 were detected in *A. chevrieri*, *A. ilex*, and *R. norvegicus*, which again suggested that the A lineage of β-CoV had intestinal tropism. Another cluster of α-CoVs, namely Poland *Myodes glareolus* 1 (PLMg1), United Kingdom *Microtus agrestis* 1 (UKMa1), United Kingdom *Microtus agrestis* 2 (UKMa2) and United Kingdom *Rattus norvegicus* 1 (UKRn1), in one of the lineages of α-CoV were only detected in liver samples of Norway rats, the bank vole, the wood mouse, and the noncyclic field vole, suggesting that they are hepatotropic [19]. In the last 30 years, a number of cross-species transmission events of CoVs, as well as changes in viral tropism, have led to major new animal and human diseases involving bovine coronavirus (BCoV), HCoV-OC43, HCoV-229E, canine coronavirus (CCoV), feline coronavirus (FCoV), porcine coronavirus (PCoV), transmissible gastroenteritis virus (TGEV), and also the recently emerged severe SARS-CoV-2 [37,38,39,40,41,42,43,44]. While SARS-CoV-2 has emerged as the most recent example of zoonotic virus spillover to humans, studies have shown that SARS-CoV-2 also has widespread tissue tropism [38]. The ability of CoV to cross species barriers and gradually spread to host animals in close contact with humans highlights the need to characterize small mammalian infections with coronaviruses, and it is also unlikely that SARS-CoV-2 will be the last CoV to cross species barriers and infect humans and other animal species [45]. However, in this study, the viral load data for other tissues were only obtained by qRT-PCR, and the gene sequence identification of the coronavirus was not carried out, except for rectal tissue. Therefore, this part of the data still has limitations.

Viral surveillance in animal reservoirs is an important step in understanding the exposure of humans to potential zoonoses, the types of human–animal interaction that impact the potential for spillover infection, and the factors that determine the transmissibility and pathogenicity of viral zoonoses in humans [46]. Whether small mammals infected with CoVs have any impact on human health and life remains to be further studied. Our current understanding of the diversity of viruses carried by small mammals, the host range of viruses, and the drivers and specific mechanisms of cross-species transmission of viruses to humans is still shallow, limiting our in-depth study of pathogens; thus, research in this area needs to be strengthened and requires special attention.

## 5. Conclusions

In this study, CoVs were detected in the six species, indicating that CoVs have a wide range of hosts. By RT-PCR and qRT-PCR, it was found that small mammals in residential areas, arable areas, and wild bush areas in Nujiang and Dali prefectures were infected with α-CoV and β-CoV. A comparison of the partial RdRp gene and phylogenetic analysis showed the genetic diversity of α-CoV and β-CoV in small mammals from the two prefectures in China, illustrating the high susceptibility of natural hosts to CoVs. Therefore, it is necessary to strengthen the monitoring of coronaviruses in Dali and Nujiang prefectures.

The qRT-PCR method based on a Taqman probe that was designed for the detection of α-CoV and β-CoV in small mammalian samples had good sensitivity, specificity, stability, and reproducibility. It was found that the detection rate of qRT-PCR was significantly higher than that of RT-PCR (*p* < 0.01). Using this method, we found that rectal tissue contained the highest number of CoV copies (*p* < 0.0001) in the detected tissues. Establishing this method not only enables rapid epidemiological investigation but also helps to provide scientific support for the study of the epidemiology and pathogenesis of CoVs infecting small mammals.

## Figures and Tables

**Figure 1 viruses-15-01965-f001:**
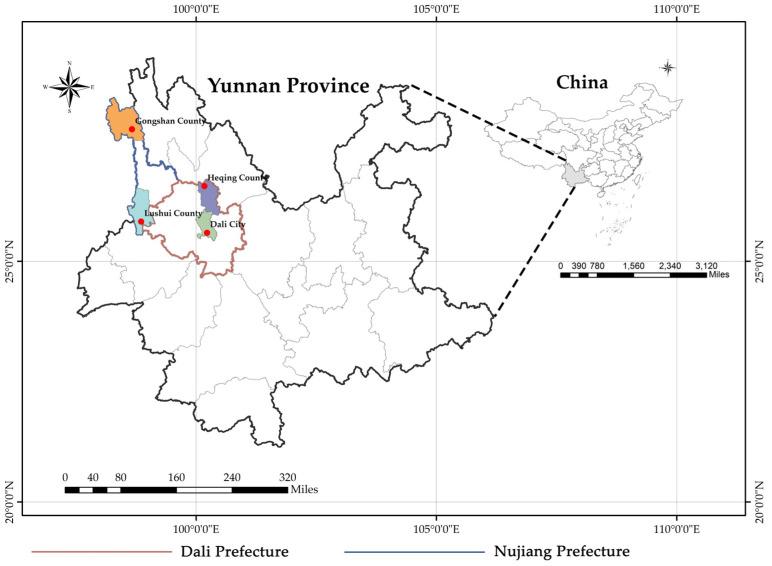
Sampling location. The right figure is a map of China, the left figure is a map of Yunnan Province, and the red dots in the figure represent the sampling points of this study.

**Figure 2 viruses-15-01965-f002:**
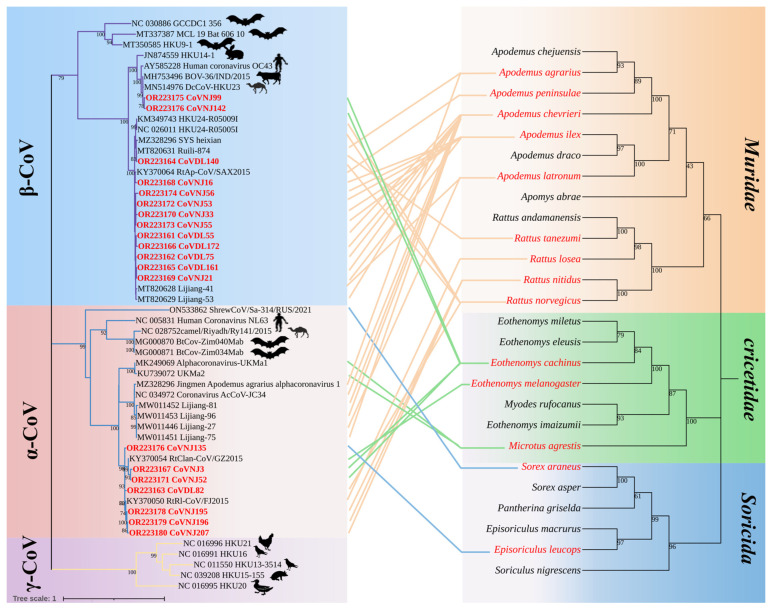
Co-evolution between coronaviruses and their hosts. The left is the phylogenetic tree established by CoV RdRp fragments, in which the virus strain marked in red is the sequence obtained in this study, and the hosts other than *Muridae*, *Cricetidae*, and *Soricidae* are labeled; the right is a phylogenetic tree constructed from the host mt-*Cytb* gene.

**Figure 3 viruses-15-01965-f003:**
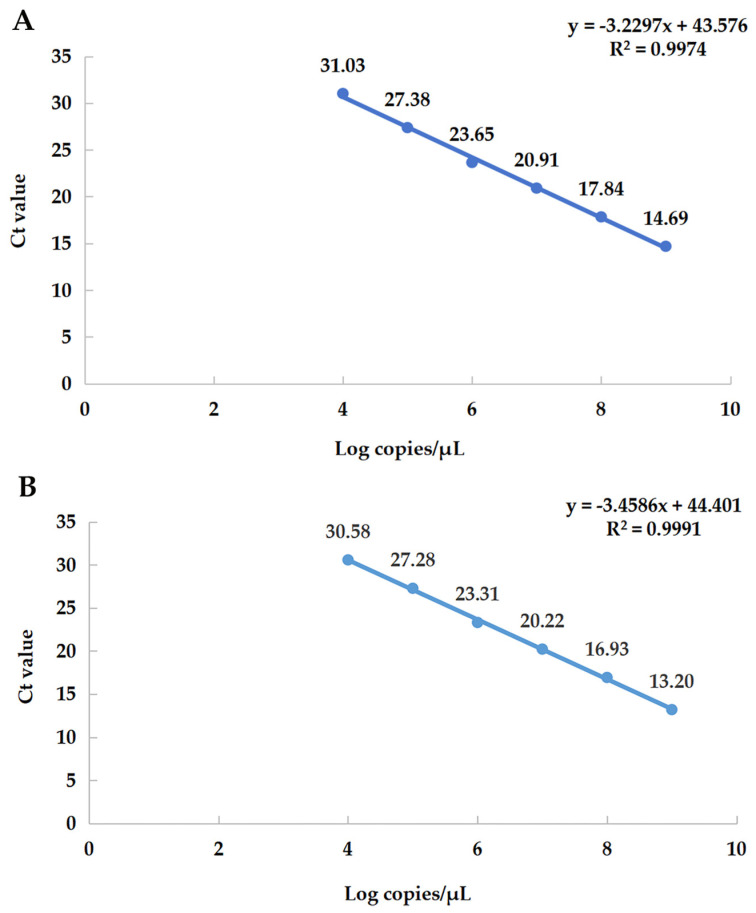
Standard curve. (**A**) The equation of the standard curve of α-CoV is y = −3.2297x + 43.576, the correlation coefficient R^2^ = 0.9974, the slope = −3.2297, and the amplification efficiency (E%) = 104%. (**B**) The equation of the standard curve of β-CoV is y = −3.4586x + 44.401, the correlation coefficient R^2^ = 0.9991, the slope = −3.4586, and the amplification efficiency (E%) = 95%.

**Figure 4 viruses-15-01965-f004:**
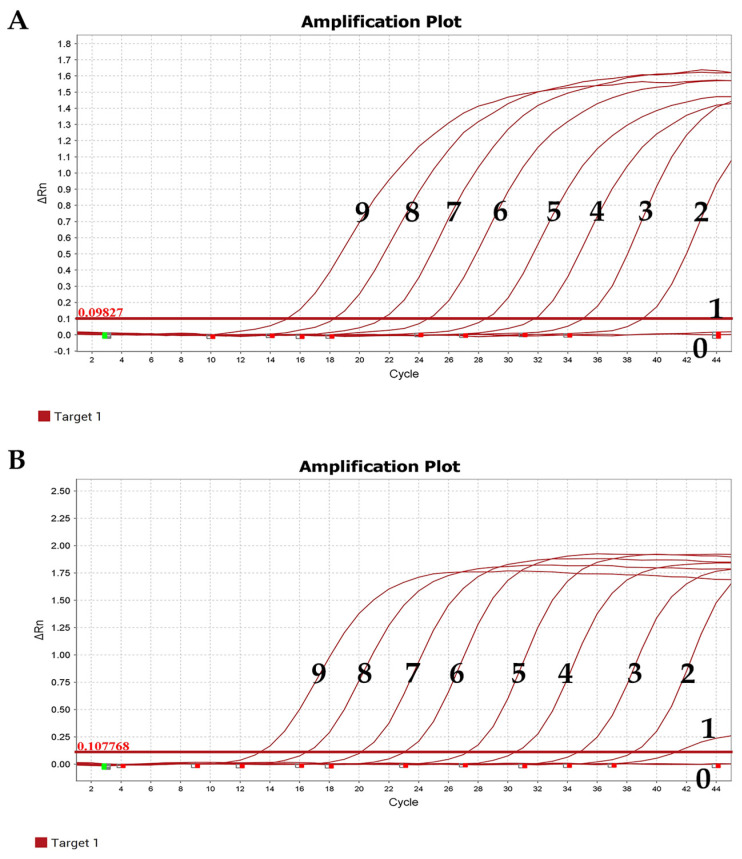
Sensitivity test results. (**A**) α-CoV. (**B**) β-CoV. 9–1: Copies: 1.00 × 10^9^–1.00 × 10^1^ copies/µL; 0: negative control.

**Figure 5 viruses-15-01965-f005:**
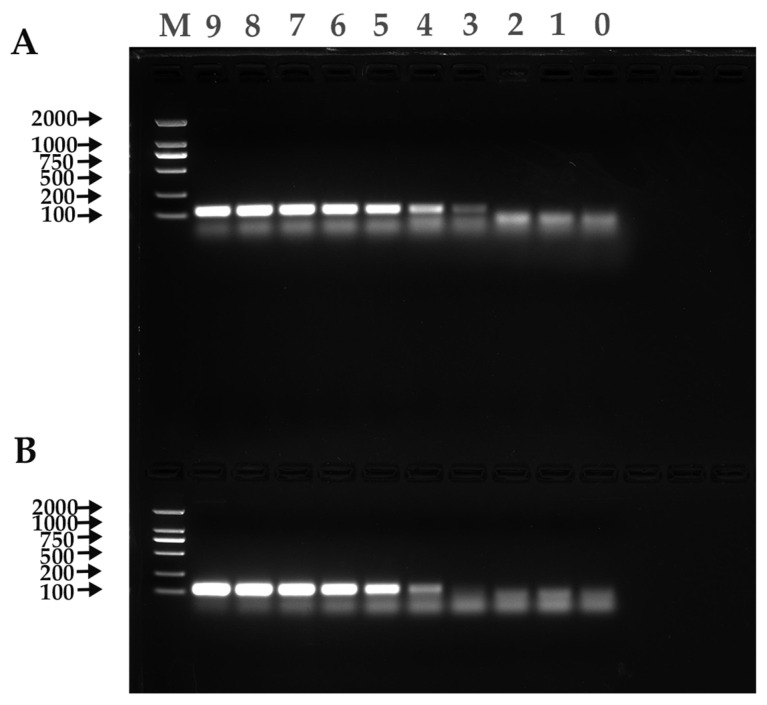
RT-PCR results. (**A**) α-CoV. (**B**) β-CoV. M: Trans2K DNA Marker; 9–1: 1.00 × 10^9^–1.00 × 10^1^ copies/µL; 0: negative control.

**Figure 6 viruses-15-01965-f006:**
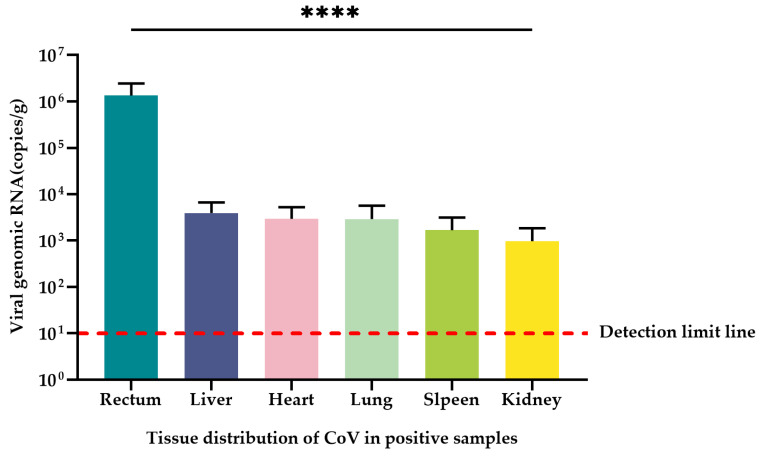
Tissue distribution of CoV in positive samples. The quantification (mean ± standard error) of CoV RNA in the rectum, liver, heart, spleen, lung, and kidney of 20 CoV-positive samples was measured in copies/g, and the significance test was performed using one-way ANOVA (**** *p* < 0.0001). The detection limit of qRT-PCR is a red dashed line in this study.

**Table 1 viruses-15-01965-t001:** The primer information in this study.

	Primer Name	Sequence (5′→3′)	bp	Amplify the Region	References
CoV RT-PCR primer	CoV-FWD3	GGTTGGGAYTAYCCHAARTGTGA	434 bp	RdRp	[23]
	CoV-FWD4/other	GAYTAYCCHAARTGTGAUMGWGC			
	CoV-RVS3	CCATCATCASWYRAATCATCATA			
α-RCoV qRT-PCR primer and probe	α-RCoV-F14493	ACATCTGGTGATGCTAGTAC	110 bp	RdRp	This study
	α-RCoV-R14602	TTCCTRCAAACATTACTATCAACAG			
	α-RCoV-Probe	FAM-TTTTCAGGCTGTTAGTGCTAATGTAAATAAATTGC-BHQ1			
β-RCoV qRT-PCR primer and probe	β-RCoV-F15324	AGTATGATGATTTTGAGTGATGATGGYGTTG	117 bp	RdRp	This study
	β-RCoV-R15440	CACGTTATTTTGATAATACAGCACCTGTTG			
	β-RCoV-Probe	FAM-TATGCGTCCAAAGGTTATATTGCTAATATTAGTGCCT-BHQ1			
Rodent identification primer	L14724	ATGATATGAAAAACCATCGTTG	1200 bp	mt-*Cytb*	[22]
	H15915	TTTCCNTTTCTGGTTTACAAGAC			

**Table 2 viruses-15-01965-t002:** The situation of CoV infection in small mammals in Dali and Nujiang prefectures, Yunnan Province.

Order	Species	Locations	Composition, %	Prevalence, %		
				qRT-PCR	RT-PCR	
					α-CoV	β-CoV
*Rodentia*	Chevrieri’s field mouse (*Apodemus chevrieri*)	Dali, Nujiang	22.51 (113/502)	9.73 (11/113)	0 (0/113)	3.54 (4/113)
	Lancangjiang field mouse (*Apodemus ilex*)	Dali, Nujiang	17.93 (90/502)	21.11 (19/90)	0 (0/90)	6.67 (6/90)
	Kachin red-backed vole (*Eothenomys cachinus*)	Dali, Nujiang	16.14 (81/502)	11.11 (9/81)	2.22 (2/90)	2.22 (2/90)
	Large oriental vole (*Eothenomys miletus*)	Dali, Nujiang	7.77 (39/502)	12.82 (5/39)	0 (0/39)	0 (0/39)
	Asian house rat (*Rattus tanezumi*)	Dali, Nujiang	14.94 (75/502)	25.33 (19/75)	0 (0/75)	0 (0/75)
	Norway rat (*Rattus norvegicus*)	Dali	5.18 (26/502)	46.15 (12/26)	3.85 (1/26)	3.85 (1/26)
	Grey bellied mouse (*Niviventer eha*)	Nujiang	0.6 (3/502)	0 (0/3)	0 (0/3)	0 (0/3)
	Ryukyu mouse (*Mus caroli*)	Dali	0.6 (3/502)	0 (0/3)	0 (0/3)	0 (0/3)
	House mouse (*Mus musculus*)	Dali	0.40 (2/502)	0 (0/2)	0 (0/2)	0 (0/2)
	Chestnut white-bellied rat (*Niviventer fulvescens*)	Dali, Nujiang	0.40 (2/502)	0 (0/2)	0 (0/2)	0 (0/2)
	White-footed Indochinese rat (*Rattus nitidus*)	Dali	0.80 (4/502)	75 (3/4)	75 (3/4)	0 (0/4)
	Swinhoe’s striped squirrel *(Tamiops swinhoei*)	Dali	0.20 (1/502)	0 (0/1)	0 (0/1)	0 (0/1)
*Insectivora*	Chinese mole shrew (*Anourosorex squamipes*)	Nujiang	4.18 (21/502)	4.67 (1/21)	0 (0/21)	0 (0/21)
	Long-tailed red-toothed shrew (*Episoriculus leucops*)	Nujiang	3.39 (17/502)	5.88 (1/17)	5.88 (1/17)	0 (0/17)
	Asian gray shrew *(Crocidura attenuata*)	Nujiang	2.19 (11/502)	27.27 (3/11)	0 (0/11)	0 (0/11)
	House musk shrew (*Suncus murinus*)	Dali	0.60 (3/502)	66.67 (2/3)	0 (0/3)	0 (0/3)
*Lagomorpha*	Tibet pika (*Ochotona thibetana*)	Nujiang	1.79 (9/502)	11.11 (1/9)	0 (0/9)	0 (0/9)
*Scandentia*	Tree shrew (*Tupaia belangeri*)	Dali	0.40 (2/502)	50 (1/2)	0 (0/2)	0 (0/2)
	Total		100 (502/502)	17.33 (87/502)	1.39 (7/502)	2.59 (13/502)

**Table 3 viruses-15-01965-t003:** CoV copies in tissue of positive samples.

	Rectum	Liver	Heart	Spleen	Lung	Kidney
Mean (copies/g)	1.35 × 10^6^	3.95 × 10^3^	2.96 × 10^3^	2.93 × 10^3^	1.68 × 10^3^	0.97 × 10^3^
SEM	1.09 × 10^6^	2.71 × 10^3^	2.28 × 10^3^	2.77 × 10^3^	1.47 × 10^3^	0.87 × 10^3^

## Data Availability

The datasets analyzed during the current study are available from the corresponding authors on reasonable request. All the sequences in this manuscript can be obtained from the NCBI database (https://www.ncbi.nlm.nih.gov, accessed on 7 July 2023).

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
