# Peer review of "Detection of Alpha- and Betacoronaviruses in Small Mammals in Western Yunnan Province, China"

_viruses, 2023, doi:10.3390/v15091965_

Round 1
Reviewer 1 Report
I read with interest this paper investigating Coronavirus detection in small mammals in two regions of China. The data presented gives a good indication of the prevalence of CoVs in the species collected and, ustilising phylogenetic analysis, situate the viruses identified within the context of published sequences, and relative to the hosts. However, the results section lacked clarity in defining the important results obtained and a table of the 20 positive samples detailing location, host species, virus detected etc is essential in order for the reader to understand how each viral sequence relates to the animal it came from. Furthermore, a localised map detailing where the samples have come from is strongly recommended to enable the reader to identify the links the authors elude to in the discussion.
In general the paper was adequately written, but the introduction was too lengthy, reading more like a review at times. This should be reduced. I am hesitant to ask for additional information in the introduction given its length, but once reduced, it would be useful to have a sentence or 2 on the reason for choosing the RdRp gene for sequencing and a rationale behind developing your own real-time RT-PCR when there are plenty already available. Also why were the species of rodents chosen (opportunistic or deliberate).
Below are a number of specific comments/questions that the authors need to address:
1. In Abstract Ln 18: latin species name should be in full: A. ilex
2. In Introduction Ln 61: typo ‘inclduing’
3. M&M Ln112: reword this sentence: ‘The samples trapping was performed using cage-type traps to capture, and the captured samples were collected at the destination early the next morning.’
4. Results Ln 263: add n= to the numbers in brackets. ‘The overall positives of CoV was 20 including β-CoV (13) and α-CoV (7) with 3.98% prevalence in intestinal tissue samples.’
5. Table 2 please add common name as well as species name to help reader.
6. Results section 3.2 and 3.3. These sections are almost impossible to read and take in. Suggest a map with each sample located colour coded by similarity. The sections should ideally summaries the important differences/main findings not just a description of the results.
7. Results sections: need a table (supplementary) which lists the (n=20?) positive samples obtained in this study alongside the published sequences used in the analysis individually with the appropriate details (collection date, location, host species and Genbank accession number).
8. Check spelling of one-humped camel (suggest its changed to dromedary camel) in results section
9. Discussion Ln 422: Clarify which CoV you are talking about: ‘Although only partial fragments were obtained in this experiment, this CoV was detected in rats in Nujiang Prefecture, Yunnan Province, China, and the related research should be strengthened.’
10. Conclusion: The first paragraph of the conclusions contains data more relevant to the results.
Some typos and strange sentences see comments above.
Author Response
Dear Reviewer 1
Thank you very much for your valuable suggestion, we have carefully revised the manuscript, please see the attachment (Revisde and Table S5) . Here is our response to your proposed modification.
- In Abstract Ln 18: latin species name should be in full: ilex
Response 1: Removed some sentences, please see the text.
In Introduction Ln 61: typo ‘inclduing’
Response 2: Changed it into “including”, please see the text.
- M&M Ln112: reword this sentence: ‘The samples trapping was performed using cage-type traps to capture, and the captured samples were collected at the destination early the next morning.’
Response 3: This sentence has been already rewritten, please see the text.
- Results Ln 263: add n= to the numbers in brackets. ‘The overall positives of CoV was 20 including β-CoV (13) and α-CoV (7) with 3.98% prevalence in intestinal tissue samples.’
Response 4: Added n=13, n=7, please see the text.
- Table 2 please add common name as well as species name to help reader.
Response 5: Added Common name, please see Table 2.
- Results section 3.2 and 3.3. These sections are almost impossible to read and take in. Suggest a map with each sample located colour coded by similarity. The sections should ideally summaries the important differences/main findings not just a description of the results.
Response 6: We reorganised the results section 3.2 and 3.3 as suggested, using graphs with different coloured samples of the originating host animals for easier understanding. Please see the results section 3.2, 3.3 and Fig 2.
- Results sections: need a table (supplementary) which lists the (n=20?) positive samples obtained in this study alongside the published sequences used in the analysis individually with the appropriate details (collection date, location, host species and Genbank accession number).
Response 7: Added supplementary (table S5).
- Check spelling of one-humped camel (suggest its changed to dromedary camel) in results section
Response 8: Revised, please see the results.
- Discussion Ln 422: Clarify which CoV you are talking about: ‘Although only partial fragments were obtained in this experiment, this CoV was detected in rats in Nujiang Prefecture, Yunnan Province, China, and the related research should be strengthened.’
Response 9: Clarified CoVs as CoVNJ99 and CoVNJ142. Please see the discussion.
- Conclusion: The first paragraph of the conclusions contains data more relevant to the results.
Response 10: The conclusion has been rewritten in order to summarize the relevant results. Please see the conclusion.

Reviewer 2 Report
Dr. Zhang team reported the detection of alpha- and Betacoronaviruses in small mammals in Western Yunnan Province. The study reported is about the small animals’ surveillance for the coronavirus exposure and viral tropism. The overall finding is interesting but can be presented in an improved way.
As most experiments, there are limitations in the reported study. The major comments are:
1. The report does not mention detail of the sites of the animal collected. How close were the collected animals to human population? the sites are all urban, suburban, rural, or remote like in the forest? This information indicates how easy the viruses can expose to general human population.
1. the animals were collected by traps with only one type of bait, that may limit the type of animals can be collected. The physical condition of the animals should also be stated (looked health or sick…).
2. The study major data is based on qRTPCR which is limited to what primers and probe used. The report should state the limitation of that clearly.
3. The study use the word “viral load” to describe their finding, but the data shown is actually viral gene detection, not infectious viruses load (by plaque assay or by TCID50 type assay) quantification, not full-length genome detection (by NGS), but just viral RNA fragment detection.
4. To validate the sensitivity of the viral gene recovery and qRTPCR assay, it is better to spike in different known amount of index viruses into the tissues collected following by evaluating the sample collection, purification, storage and PCR-based quantification procedure.
5. If the sera samples of the collected animals are still available, it may be interesting for knowing if the animal had exposed or remained serologically positive to the selected coronaviruses.
6. The manuscript should state the limitations of the study and the future research direction in the discussion.
Minor comments:
1. The abstract is too long.
2. Line 101, tone-down the word “absolutely”
3. Line 125, which part of the intestine tissue were collected in the reported study? Small intestines, duodenum, ileum, large intestine, caecum colon or rectum?
4. Line 409 to 412, use “camel” instead of “camelus”.
3. Figure 1, more detail map of the animal collected sites.
4. Figure 2, The number of samples should be next to the host order [ like Murida (424)…] to indicated the size of the sample in each hosts. The lines connecting the three indicated hosts to the CoVs should in three different colors to clarify the point made.
5. For the tropism experiment, brain tissue and muscle tissue were collected?
6. Figure 6. Detection limit line should be included in the figure.
7. Table 2. A column stating the common names of the animal collected should be there.
Some sentences are too long.
Author Response
Dear Reviewer 2
Thank you very much for your valuable suggestion, we have carefully revised the manuscript, please see the attachment (Revisde and Table S5) . Here is our response to your proposed modification.
- The abstract is too long.
Response 1: Revised, please see the abstract.
- Line 101, tone-down the word “absolutely”
Response 2: “absolutely” was removed, please see the text.
- Line 125, which part of the intestine tissue were collected in the reported study? Small intestines, duodenum, ileum, large intestine, caecum colon or rectum?
Response 3: We collected the rectal tissue. Replaced all intestinal tissue with rectal tissue in the article.
- Line 409 to 412, use “camel” instead of “camelus”.
Response 4: Revised, please see the text.
- Figure 1, more detail map of the animal collected sites.
Response 5: Revised, please see Figure 1.
- Figure 2, The number of samples should be next to the host order [ like Murida (424)…] to indicated the size of the sample in each hosts. The lines connecting the three indicated hosts to the CoVs should in three different colors to clarify the point made.
Response 6: Revised, please see Figure 2.
- For the tropism experiment, brain tissue and muscle tissue were collected?
Response 7: No, the brain tissue and muscle tissue have been not collected.
- Figure 6. Detection limit line should be included in the figure.
Response 8: Revised, please see Figure 2. The detection limit line of qRT-PCR is a red dashed line in Figure 2.
Table 2. A column stating the common names of the animal collected should be there.
Response 9: Added the common names in Table 2.

Reviewer 3 Report
General comments
The manuscript by Xu et al. describes the detection of alpha and beta coronaviruses in small mammals in Western Yunnan province, China. To this end, RT-PCR detection of partial RdRp gene of CoVs were determined in more than 500 small mammals from 18 species in 12 genera collected from two prefectures of Western Yunnan province in China. The prevalence of beta-CoVs ranged from 3,54% to 6.67 in rats and mice, respectively.
The identity of partial RdRp genes of 13 strains of beta-CoVs was up to 99.23% at the nucleotide level. Very high sequence identity up to 94% was detected in mammals: Indian bovine and camels in Marroco. This and complementary data showed the genetic diversity of alpha- and beta-CoVs in small mammals from the two prefectures in China, illustrating the high susceptibility of natural hosts to CoVs.
The manuscript includes a vast amount of data on the dissemination of CoVs in mammals, providing interesting information. Of course, it would deeply reinforce the interest of the manuscript the inclusion of data on the infection of humans with CoVs to determine whether they retain RdRp sequence identity among them and the extent of species crossing by these viruses. Still, the manuscript is the high interest to a wide audience of scientist.
Specific points
1. Line 41. Authors quote that coronavirus genome of 30 000 nt is the largest for an RNA virus, which is correct, nevertheless, they may want to quote that there are other nidovirus with a genome of 35.5 kb: (https://doi.org/10.1371/journal.ppat.1007314 November 1, 2018 1 /)
2. Figure 1. In the description of the figure it is indicated that on the left side a map of China is shown, whereas on the right a map of Yunnan Province. Nevertheless, it seems that it is the other way around.
3. The manuscript describes a vast amount of interesting information, but the precise information provided is overwhelming. This reviewer suggests to reduce the amount of specific data provided in the manuscript and to provide summaries of this information, moving some of the details to complementary information figures or tables.
4. Lines 278-301. Comparison of partial RdRp gene nucleotide identity. Similar to what has been indicated in point 3, in this section very high identities of RdRp from different viruses isolated in China are shown. Unfortunately, the density of precise data does not help reading of the manuscript. This reviewer would suggest again to summarize the message of this section by including the specific data in supplementary tables.
5. 5. Fig. 2. In this figure the coevolution between coronaviruses and their host is described. Unfortunately, no conclusion or comment on the infection of different mammals by the different CoVs: alpha, beta or gamma is provided. Is there any suggestion for the possible relationship?
6. 6. Lines 370-380 and Fig 6. Tissue tropism. This reviewer finds very interesting the observation on the higher abundance of the CoVs in the intestinal tract (one thousand-fold higher). Authors may want to expand this result indicating which could be the reason for this interesting observation.
Minor points
Line 44. Space needed between
Acceptable, but it could be improved
Author Response
Dear Reviewer 3
Thank you very much for your valuable suggestion, we have carefully revised the manuscript, please see the attachment (Revisde and Table S5) . Here is our response to your proposed modification.
- Line 41. Authors quote that coronavirus genome of 30 000 nt is the largest for an RNA virus, which is correct, nevertheless, they may want to quote that there are other nidovirus with a genome of 35.5 kb: (https://doi.org/10.1371/journal.ppat.1007314 November 1, 2018 1 /)
Response 1: We deleted: which is the largest RNA genome of virus discovered so far.
- Figure 1. In the description of the figure it is indicated that on the left side a map of China is shown, whereas on the right a map of Yunnan Province. Nevertheless, it seems that it is the other way around.
Response 2: Revised, please see Figure 1.
- The manuscript describes a vast amount of interesting information, but the precise information provided is overwhelming. This reviewer suggests to reduce the amount of specific data provided in the manuscript and to provide summaries of this information, moving some of the details to complementary information figures or tables.
Response 3: The suggestion is very good. We will summarize some information in the Supplementary Materials (Table S5).
- Lines 278-301. Comparison of partial RdRp gene nucleotide identity. Similar to what has been indicated in point 3, in this section very high identities of RdRp from different viruses isolated in China are shown. Unfortunately, the density of precise data does not help reading of the manuscript. This reviewer would suggest again to summarize the message of this section by including the specific data in supplementary tables.
Response 4: The suggestion is very good. We will summarize some information in the Supplementary Materials (Table S5).
- Fig. 2. In this figure the coevolution between coronaviruses and their host is described. Unfortunately, no conclusion or comment on the infection of different mammals by the different CoVs: alpha, beta or gamma is provided. Is there any suggestion for the possible relationship?
Response 5: Revised: From the α-CoV and β-CoV and host co-evolutionary tree, it was found that CoV of different genera could infect the same rodents and CoV of the same genera could infect different rodents, suggesting the existence of cross-species transmission of α-CoV and β-CoV carried by rodents inhabiting the same habitat. It is speculated that the result of long-term evolution, mutual adaptation, and natural selection between CoVs and host animals.
- Lines 370-380 and Fig 6. Tissue tropism. This reviewer finds very interesting the observation on the higher abundance of the CoVs in the intestinal tract (one thousand-fold higher). Authors may want to expand this result indicating which could be the reason for this interesting observation.
Response 6: Thank you for your suggestion. Added:In the relevant research on SARS-CoV-2, some researchers have found that the faeces remained positive in 23% of patients even after respiratory specimens tested negative for viral RNA, it is speculated that the gastrointestinal tract may be a specific target organ of the virus [29]. In previous studies on SARS-CoV, it was found that, prolonged faecal shedding of viral RNA is common, and faecal samples remain positive even after the respiratory and/or sputum samples exhibit no detectable virus [30]. Therefore, there were also traces of the highest average CoV load in rectal tissue in this study.
Minor points
Line 44. Space needed between
Response: Revised, please the text.
